# Animal Welfare Considerations and Ethical Dilemmas Inherent in the Euthanasia of Blind Canine Patients

**DOI:** 10.3390/ani12070913

**Published:** 2022-04-02

**Authors:** Vito Biondi, Michela Pugliese, Eva Voslarova, Alessandra Landi, Annamaria Passantino

**Affiliations:** 1Department of Veterinary Sciences, University of Messina, 98168 Messina, Italy; vbiondi@unime.it (V.B.); passanna@unime.it (A.P.); 2Department of Animal Protection and Welfare and Veterinary Public Health, Faculty of Veterinary Hygiene and Ecology, University of Veterinary Sciences Brno, 612 42 Brno, Czech Republic; voslarovae@vfu.cz; 3Veterinary Practitioner, 10129 Turin, Italy; alelandi1612@gmail.com

**Keywords:** ocular disease, blindness, good death, well-being, veterinary ethics

## Abstract

**Simple Summary:**

Although many dogs with blindness diagnosis can reach a similar age compared to those not affected, often the owners require euthanasia of their animals. This choice leads to conflicting moral principles relating to what is better for the animal and the owner. This article discusses the suitability of euthanasia in blind dogs. To better assess factors influencing the choice of euthanasia, four different scenarios were constructed that described various situations regarding the animal’s aptitude, pet owner, and veterinarian relations.

**Abstract:**

In dogs, several primary or secondary diseases affecting the ocular structures may cause blindness. In cases where the visual impairment is not associated with severe systemic involvement and the animal can still have, predictably, a good “long-term” quality of life, the veterinarian should inform the owner about the differences between humans and animals, concerning the type of visual perception. In the light of the daily findings in veterinary clinic practice, the Authors report four different scenarios with conflicting views between veterinarians and owners about the euthanasia request for a blind dog. They underline how the diagnosis of incipient or already established blindness in dogs can sometimes lead to an inappropriate request for euthanasia.

## 1. Introduction

Dogs’ blindness often is due to a variety of acute and progressive diseases or trauma or can be consequent to Brachycephalic Ocular Syndrome [1]. Some of the most common causes of vision loss in dogs include cataracts, glaucoma, sudden acquired retinal degeneration syndrome (SARDS), progressive retinal atrophy (PRA), and retinal detachment (RD). Blindness can be also caused by certain cancers (brain and orbital tumors), neurologic diseases, and drug overdoses that affect the retina (e.g., an overdose with ivermectin) [2].

The main causes of blindness are reported in Table 1.

Loss of vision can occur suddenly or develop gradually over time. It may be complete (involving both eyes) or partial, involving only one eye or even certain parts of the visual field. To best treat patients and maintain their good “long-term” quality of life, a complete clinical examination of the dog considering potential clinical manifestations of the diseases involving the eyes is required. Indeed, many diseases such as diabetes, Cushing’s syndrome or hypertension show serious ocular signs that may compromise the quality of life.

Identifying the specific etiology of blindness and/or ocular structures involved is fundamental to the veterinarian advising owners on the prognosis for the animal’s vision and general health [4]. Then, it is necessary to carry out a diagnostic approach that includes:(i)an ophthalmic history identifying the onset and duration of blindness, degree of blindness (as perceived by the owner that could often report disorientation, clumsiness, and/or anxiety of the dog) [5], if there have been any changes in the appearance of the eyes and any behavior changes, other signs of disease, and received treatments.(ii)vision assessment, using the menace response test, the visual placing reaction, and the maze test [6,7];(iii)causative lesion localization by Pupillary Light Reflex (PLR) examination of the eye, potentially with ocular ultrasound, electroretinogram (ERG), and/or functional magnetic resonance imaging (MRI) [8].

Therefore, the veterinarian must help the owner to comprehend why his/her dog has developed a loss of vision (whether the blindness is complete or partial), if systemic signs of disease are present or systemic diseases have been previously diagnosed (because often several systemic diseases, such as infectious diseases, hypertension, etc., may initially be recognized by their ocular manifestations), and how to help the animal to adapt to ocular impairment. Furthermore, modern progress in veterinary ophthalmic surgery, such as cataract removal by phacoemulsification, retinal detachment surgery, and laser endoscopic cyclophotocoagulation, has increased the treatment options for the patients that previously would be permanently blind [9,10,11,12], thereby improving their life quality. Although many dogs with blindness diagnosis can live a good life [13], the owners require euthanasia of the animal but with a treatable and/or non-fatal condition (“convenience euthanasia”), either for financial reasons or simply because they do not want he/she anymore. This choice raises ethical questions because of veterinarians’ duties towards both animals and owners, particularly if owners’ wants are against the animal’s interests or welfare. This is because the animals, by biocentric viewpoint, are viewed as moral patients and their interests are considered to be a priority; veterinarians act as advocates to protect them, tending to act similar to pediatricians [14]. This objective is in contrast with convenience euthanasia [15]. Furthermore, lacking a clearly defined set of rules when approaching a canine blindness case could result in the development of conflicting moral principles. Thus, decisions over dog euthanasia must be made considering the severity of the condition, the long-term prognosis, the animal’s rights, and the interests of the owners. The veterinarian should weigh up all of these considerations before deciding whether euthanasia is the best course of action.

Although there are a variety of potential scenarios that involve the euthanasia dilemma in daily clinical practice, in this study the focus is to discuss if putting down a dog with complete blindness is right.

To better discuss the ethical dilemmas inherent in the decision-making around euthanasia of a blind canine patient, four different hypothetical scenarios that describe realistic clinical cases relating to dogs with the diagnosis of blindness were constructed.

## 2. Blindness from the Animal Welfare Point of View

### 2.1. Animal Welfare and Ethics

All animals, regardless of the species to which it belongs, are considered sentient beings capable of feeling emotions such as pleasure, pain, stress, and suffering, both at physical and psychological levels. Animal welfare has been defined in several different ways [16,17,18,19,20,21,22,23,24,25]; it can be considered a positive mental and physical condition related to the satisfaction of its physiological and behavioral needs and its expectations [26]. This state varies according to the animal’s perception of its situation. Animal welfare includes three concepts: (i) an animal’s feelings; (ii) an animal’s ability to perform natural behavior, and (iii) an animal’s health and biological functioning [27,28]. To evaluate animal welfare, all of these concepts (behavior, affective state, and health and biological functioning) should be considered.

Good animal welfare requires disease prevention and veterinary treatment, appropriate management, nutrition, responsible care, and, when necessary, a humane death [28].

Although welfare scientists have not always agreed on the animal welfare definition [29], our view is that good health and the absence of stress is not enough to ensure welfare. It is also necessary to consider what the animal feels, not only unpleasant subjective perceptions, as mentioned above, but also to look for the signs of expression of positive emotions [30,31,32,33]. The analysis of behaviors and health status of the animal provides an integrated vision of its welfare [34].

There are several pro-social and behavior signals of welfare in dogs indicating a positive emotional state such as relaxed body posture and facial expression (ears to the side or forward if anticipating a positive event), consuming food when offered, playing, soliciting attention, etc. Instead, some medical problems can modify animal behavior. A medical condition such as blindness can, however, modify animal behavior as it may modify or prevent the animal’s perception of the environment. Partial blindness could cause fears or phobias [35]. If the ocular problem causes pain, the animal changes behavior (decreased interaction with other animals or humans, reduction of appetite, aggression, fear reaction, vocalization, altered facial expression, and/or posture, restlessness, etc.) [36].

Sharing the thought of Rollin [37], in our opinion, a significant part of animal welfare is ethics, given that it influences an individual’s concept of welfare, even if other Authors believe that animal welfare science is not interchangeable with or a synonym of animal ethics [38]. In this context, ethics would prescribe the veterinarian’s course of action in terms of the blind animal’s welfare as well as the rights/obligations/welfare of the owner. In fact, the veterinarian has an ethical obligation to provide good care for his/her patient. Ethics may also be considered when addressing value-based questions such as the acceptability of animal quality of life. Therefore, the veterinarian should take into consideration the animal’s needs and discuss potential ethical concerns with the owner, taking the role of patient advocate [39]. In this way, his/her actions would primarily look after the well-being of the animal following the paediatrician model based on the moral value of the animal [15].

### 2.2. Blindness and Animal Welfare

Given that poor quality of life as sensed by the owner implies a reason for euthanizing the companion animals [40,41], management of any disease that is blinding is important both from an animal welfare point of view (concerning, e.g., the pain) and from an owner’s viewpoint (e.g., respect of the treatment).

Given that the perception of pain decreases the quality of life in all animals, respect for animal welfare and dignity must be viewed as the main principle in the veterinary practice, with the consequent need for a better understanding of animal pain management from an ethical viewpoint. Alleviating pain is necessary for both ethical and clinical reasons [42]. Animals are a valued part of society and must be protected from needless suffering [43]. From a clinical standpoint, the pain has detrimental effects on animal well-being [42], and untreated pain can lead to weight loss, aggression, self-mutilation, and heightened pain sensitivity. As above mentioned, the issue of persistent pain in any blind eye is a serious animal welfare concern, though not all blindness conditions are painful. In estimating the deleterious effect of blindness in a dog, it is important to pay attention to the potential pain because the animal does not always seem to demonstrate visible signs of discomfort.

Severe ocular pain is present in conditions such as endophthalmitis, glaucoma, and penetrating ocular trauma. A study reported that the chronicity of the pain associated with glaucoma exacerbated lethargy [44]. This implies that it may be a reliable indicator of the quality of life [45]. For these and other ocular diseases with painful eyes, to avoid dog euthanasia, enucleation surgery may be strongly recommended as an end-stage solution to irreversible pain. However, enucleation, especially if bilateral, might be seen as a discouraging and reluctant solution for many owners, as highlighted by owners of horses by Wright et al. (2018) [46]. A recent study investigating owners’ perceptions of their dogs’ quality of life following bilateral enucleation assessed their satisfaction with the procedure [47]. The owners showed good satisfaction and perception of improvement in their dog’s quality of life, considering this surgical procedure a viable option for managing dogs with irreversible pain. Hamzianpour et al. (2019) [47] found that a greater part of patients appeared able to return to their activity and playfulness and showed a lack of aversion to facial/ocular palpation.

In this perspective, the veterinarian must educate the owners, for example, to quickly recognize ocular pain, explain in more detail when enucleation may be necessary, and better manage the dog [44] because the management that optimizes the blind dog’s welfare depends on multiple interventions aimed at creating a safe and comfortable home environment. Sudden onset blindness could be significantly harder for both the dog and owner than a gradual loss of vision. To reduce the stress because of the progressive blindness, it could be utilized to adapt to social and physical environments and new relational practices by increasing the predictability and controllability of the animal’s situation [48]. In this regard, it might be useful to remove potentially hazardous objects that the blind dog may bump into; to leave food and water bowls in the same place; to place scent or tactile location signals to aid orientation around the house; to consider the levels of lighting, e.g., dogs with PRA function better in bright light. Welfare management strategies therefore should be considered part of a more comprehensive medical approach. Because animal welfare considers psychological as well as physical aspects, blindness can change the behavior of the animal, i.e., modifying or preventing the perception of the environment [35]. If the loss of sight is gradual, the dog compensates gradually, and behavior changes may be slight and not noticeable until the animal is completely blind. Sudden blindness can result in more dramatic behavior changes: the dog may be disoriented and hesitant when walking, bumps into things, and may vocalize more often. Dogs who become blind suddenly may also develop uncharacteristic behaviors until they learn to adapt. They may be unwilling to leave their sleeping area and may also be reluctant to explore, may appear withdrawn, and may bark when disoriented or in need of reassurance.

Therefore, to adapt living conditions to the dog’s impaired vision means guaranteeing his/her welfare.

### 2.3. Is Euthanasia in the Dog’s Interest?

Determining whether euthanasia is in a dog’s interest remains a complex and difficult matter. There is no prescriptive list evaluating when to resort to euthanasia. However, this evaluation could be done by weighing up the benefits and harms in terms of suffering and longevity of the animal, i.e., in the sense that life can continue only if beneficial—that the resultant life is worth living. Even if it is not clear whether nonhuman animals might be able to develop a desire to die in certain situations, death can be in their best interest [49]. Regarding the above evaluation, some Authors [50,51] have proposed a checklist that includes the degree of pain the animal can experience such as changes in walking, presence or absence of appetite, hydration, breathing without or with difficulty, normal urination and defecation, etc., in other words, conditions that could lead to suffering. Therefore, the incapacity to prevent any suffering could legitimize the euthanasia of any dog. Therefore, euthanasia could be justified when an animal cannot avoid having a life of overall negative welfare except by dying, and any medical over-treatment would be unethical. Indeed, owners considering their dogs as members of the family might want to utilize every possible treatment to ensure their animals remain alive if possible and probably against the interest of the animal in the case of a poor prognosis. A conflicting situation could arise between moral arguments in favor of the animal’s life and situations such as time, effort, and money that finally determine the euthanasia choice despite what is in the best interest of the animal. For this reason, as will be mentioned in the “Final Thoughts” section, it could be considered useful in euthanasia decision an algorithm that includes questions such as “Is the benefit to the owner of euthanasia greater than the harm to the animal?” [52]. Yates (2010) [52] affirmed that a relevant evaluation is not of the animal’s state in the present time, but its expected welfare over time until the next evaluation. This makes the assessment even more complicated because it involves a balance between the welfare and each of the numerous potential outcomes. This should be annealed by knowledge of human–animal differences. For example, it is not well defined whether animals have long-term objectives that they hope to realize, death, anxiety, or can predict that their suffering will finish. 

## 3. Different Scenarios Concerning the Euthanasia Request

In clinical practice, there is no single and applied framework to determine the correct approach of a request for euthanasia in blind dogs because each must evaluate the animal needs and the context in which the case is referred. To understand what ethical approach should be taken into consideration to establish when euthanasia is appropriate, we report four different realistic scenarios in which the owner bringing his/her dog with blindness (complete or partial) to the veterinarian requires euthanasia although it seemed to be not appropriate by veterinarian’s viewpoint.


*Scenario 1: Irreversible blindness and dogs’ adaption*



*A nine-year-old female pug affected with Cushing’s syndrome was presented for evaluation of rapid loss of vision developed in the last weeks. The absence of menace response and a decrease of the pupillary light response was revealed in both eyes. At the ophthalmological examination, the tapetal fundus appeared hyper-reflective, with reduction of the retinal vasculature and pallor of the optic nerve head. SARDS was suspected and confirmed from the electroretinogram test. The owner was informed of the irreversible blindness and required the euthanasia of his dog believing that she did not have a good quality of life. He also considered that the presence of frequent urination due to the syndrome was frustrating to himself.*


Irreversible blindness is a complex scenario for the emotional involvement of the owner because appropriate management of blind animals is always challenging, requiring several skills in addition to clinical knowledge. Though some conditions are irreversible, blindness could be generally well-tolerated, and dogs could adapt well with minor adjustments to their lifestyle, maintaining a good quality of life [13]. Relating to the frequent urination as reported in the above case, it can be pharmacologically treated. In a survey carried out by Stuckey et al. (2013) [53], it was reported that most owners of dogs affected with SARDS perceived the quality of their animal’s life as good, therefore discouraging euthanasia of dogs with SARDS. Though the literature suggests that blind companion animals may have an excellent capacity for adaptation, it often happens that the owners do not appreciate this, and they were distressed because their dogs presented abruptly with irreversible blindness, requiring euthanasia.

Comprehensibly, owners often do feel empathy for their animal [54] and imagine themselves being blind and the consequences it would have on their lives.

Consequently, given that dogs lack visual acuity, they adapt more easily to blindness than humans. Dogs’ vision is based more on shape, movement, and contrast than on detail [55]. Moreover, their other senses, such as hearing, touch and smell, and taste, become more highly attuned to permit them to successfully navigate to seek food, water, etc. [56,57]. Svensson et al. (2016) [58] reported that dogs with PRA learned to adapt to their decreased visual function so that the owner rarely recognized visual impairment until a late stage of the disease. Menotti-Raymond et al. (2010) [59] described the ability of cats with PRA to adapt to the progressive visual impairment using their other well-developed senses. Therefore, in our opinion, vision loss alone is not enough justification for euthanasia. For this reason, the veterinarian may refuse to euthanize, believing that dogs typically have a good quality of life despite blindness. Despite the adaptability, deciding to euthanize a blind dog may become permissible if the animal is geriatric with concomitant debilitating diseases and little hope of full recovery.


*Scenario 2: Blindness in working dogs*



*A 7-year-old male English Setter was presented to a referral hospital for a history of conjunctivitis and hesitance navigating around objects (trees, branches, and bushes.) in the last days. Menace response, dazzle reflex, and pupillary light reflex were negative, indicating blindness of both eyes. A complete ophthalmic examination, including slit-lamp biomicroscopy, tonometry, and indirect ophthalmoscopy, was performed. Conjunctivital petechiae and corneal edema in the right eye and hyphema in the left eye were observed. Ocular ultrasonography showed a complete retinal detachment and vitreous hemorrhage. Canine monocytic ehrlichiosis was diagnosed. The owner did not correctly perform the pharmacological treatment for ehrlichiosis, and the condition of the eyes worsened. The veterinarian suggested bilateral enucleation. The owner did not agree to the veterinarian’s proposal because the dog was no longer suitable for the purpose for which he was bought (hunting). Consequently, he required euthanasia.*


In other situations when owners adopt companion animals for a very specific purpose, such as guarding or hunting dogs, and the animal becomes unable to perform the desired purpose because is blind, the owner may require his/her euthanasia. This happens because companion animals are considered tangible personal property in the strict legal sense [43] and can be euthanized for any rationale their owners think up. In fact, in a recent study relating to the origin of the convenience euthanasia dilemma, it has been reported that the request of convenience euthanasia from owners is normal because animals are also seen as objects in their utility role in relation with owners, contributing to minimize needs and interests of the animal [60]. In this scenario, the owner, being disappointed in his/her animal’s expectations because of the animal being unsuitable for the purpose for which it was intended (i.e., hunting) due to the blindness, had an illusionary vision of what his/her dog should have been and done. The owner rejected his/her animal easily (that is, required euthanasia) knowing that it would be simple to find another one that perhaps met his/her expectations [60].

In these cases, when death is not in a dog’s best interest, the veterinarian, unconvinced that the burden on the owner is great enough to justify the animal’s death, must consider other reasons for euthanizing the animal. Therefore, the veterinarian must decide whether euthanizing an animal is justifiable and, whether he/she can ethically decline an owner’s request to perform euthanasia, offering to the owner alternatives and suggestions for cases in which the animal is blind, i.e., avoiding changes in the domestic environment, use auditory stimulation, etc. Conversely, if the owner is not able to correctly perform a treatment, he would not be able to support his blind dog. In this context, the choice of the veterinarian to euthanize the dog may be difficult. However, if the dog presents a good prognosis, as a medical professional, the veterinarian must act as an advocate to protect his/her patients. This is because all animals have preferences, interests, needs, and can feel pleasure and pain [61,62]. In other words, they are seen morally as having an inherent value similar to humans [62] and consequently, are entitled to have respectful treatment.

Those who possess an inherent value (humans and animals) must be treated as ends in themselves, and whenever they are treated as a means to an end (in the case of dogs used for the hunting)—as if they had a value—then they are treated with a lack of respect.

Since there is a connection between rights and duties, moral rights also generate duties not to kill physically healthy animals (blind dogs).

Nevertheless, some veterinarians prioritize the owner’s interest and proceed with euthanasia [60].


*Scenario 3. Financial constraints of the pet owner*



*A 9-year-old Labrador Retriever suffered from a mature cataract in both eyes. The veterinarian determined that the animal may be a candidate for surgery, but cataract extraction surgery by phacoemulsification required a cost of about 2500.00 €. Besides, before the surgery, the animal needed to have specific tests (i.e., electroretinogram, ocular ultrasound, comprehensive blood panel) to evaluate retinal and overall health to provide a prognosis for return of vision following the surgery, with additional costs. The owner did not want to spend the money either on the surgical treatment or the tests, but required to euthanize the animal.*


In this scenario, where a cataract impairs the dog’s vision, the owner is not willing to pay for the animal’s treatment, even if the surgical treatment such as cataract extraction surgery by phacoemulsification can improve his/her vision. Consequently, he/she decides to euthanize his/her animal. Therefore, in this economic decision making, euthanasia becomes a viable and economic alternative, although, as reported by some Authors [63], the request of euthanasia for economic limitations is one of the more frequent stressful scenarios involving the veterinarian.

Financial reasons are considered by Herfen et al. [64] as the *ultima ratio* for euthanasia. In this case, premature euthanizing a dog who became blind is done for instrumental rather than compassionate reasons.

In the case in which the owner is unable to afford treatment, it may be suitable “to make known the options and eligibility for charitable assistance or referral for charitable treatment”, as stated by the RCVS (Royal College of Veterinary Surgeons) Code of Professional Conduct for Veterinary Surgeons [65]. Another financial instrument to reduce the burden of veterinary medical costs on the owner could be pet health insurance that would consequently reduce or abolish the risk for euthanasia [66]. In our opinion, animal welfare should be the veterinarian’s primary concern. Even if animals are considered not able to participate in the decision making [67], the veterinarian does not treat them as objects (the client’s property) but as subjects, patients who deserve quality medical care. The veterinarian, where there is no legal framework about euthanasia in dogs and cats, is morally obligated to attempt influencing client decisions, although he/she also has a duty to the owner and should consider his/her autonomy in decision making [52]. It must be the veterinarian’s responsibility to use his/her judgment on different options beneficial to the patient and present all available options to the owner ranging from the best practice (gold standard) [68,69] to euthanasia (even if in some cases, e.g., in intracranial neoplasia, the best practice may be opting for euthanasia). Indeed, we believe that client education is essential, and the exchange of knowledge between veterinarian and owner is an important factor in the decision-making process [70]. Autonomy in decision making requires the owner to adequately comprehend what the situation involves. Consequently, the respect for autonomy is strictly associated with informed consent. Informed consent, in turn, may be comprehended as given only if the owner receives and understands full information on the options available and voluntarily chooses one of those options. The veterinarian must also inform the owner of uncertainties about the treatment and about what a treatment option involves. Together, respecting owner autonomy and obtaining informed consent, aims to reduce the risk of unduly influencing the patient’s decision making and provide guidance in veterinarian–client relationships.


*Scenario 4. Central nervous system (CNS) blindness*



*A ten-year-old female English cocker spaniel was presented to a referral hospital for a 24-h history of blindness associated with lethargy occurring in the last weeks. During the neurological examination, the animal showed disorientation and tended to bump into objects. The menace response was bilaterally absent, as well as the response to the cotton ball test; pupillary light reflexes were reduced, pupils appeared mydriatic, and dazzle reflex was normal. A complete blood count and biochemical tests were unremarkable. The results of neurological examination suggested the presence of a lesion localizing to bilateral retinas, optic nerves, optic tracts, or optic chiasm. The veterinarian proposed diagnostic investigations to formulate a more accurate diagnosis (Magnetic Resonance Imaging of the brain, thoracic radiographs, and abdominal ultrasound), but the owner required euthanasia.*


This scenario takes into consideration central nervous system (CNS) blindness. Several infectious and conditions causing inflammation of the CNS can affect the visual cortex and cause cortical blindness. Given that the optic nerve is an extension of the CNS, some diseases, such as granulomatous meningo–encephalomyelitis (GME) and infectious optic neuropathies due to distemper (paramyxovirus) and tick-borne encephalitis (flavivirus), can lead to central blindness. Intracranial neoplasia, of which meningioma is the most common primary tumor found in the dog [71], can also lead to sudden blindness. Although the disease process is progressive, the gradual onset of more subtle vision deficits is not recorded by owners. To increase survival time in dogs with brain tumors, it might be worthwhile for surgical resection and fractionated radiotherapy [72], although these tumors are associated with significant morbidity [73]. Rossmeisl et al. (2013) [74] reported that the prognosis for dogs with palliatively treated primary brain tumors is poor. Given that, some Authors [75] highlight how it is difficult for veterinarians confidently to advise to owners treatment decisions because there have been no robustly designed formal prospective clinical trials comparing various treatment options for intracranial tumors. They suggest that owners would know the data for all-cause mortality because they ignore that other lesions might contribute to their animal’s death within a specified period. Unfortunately, in these situations, it is not easy to be able to opt for one (treatment) or the other solution (euthanasia), given the influence of tumor-associated variables such as lesion burden or histopathologic type and grade and the severity of neurologic dysfunction [74]. For example, euthanasia could be performed due to refractory seizures or a sudden decompensation to the clinical signs existing before the treatment. Therefore, the decision to perform euthanasia or not should consider some factors of the patient animal (e.g., quality of life and advanced-stage disease), the owner (e.g., previous experience of dogs with tumors, emotional bond with the animal, and perceptions concerning tumor treatment), and veterinarian (e.g., experience and training in oncology treatment, knowledge of tumor behavior and cancer treatment options) and be taken based on the expected biological behavior of the tumor and expected benefit to the animal. Without any doubt, in our opinion, owners should be advised of the survival times based on the neuroanatomic location of the tumor when discussing outcome expectations with palliative treatments or when attempting to compare the relative efficacy of a brain tumor treatment or when choosing to euthanize the animal. The most important thing is the dog’s quality of life. A treatment that could prolong his/her life may also prolong his/her suffering.

## 4. Final Thoughts

Scenarios make us realize how under a relational logic of care, moral decisions cannot be made in the abstract sense.

To assist the pet owner in the euthanasia decision-making process, it could be useful to apply the algorithm for euthanasia decisions produced by the BVA’s (British Veterinary Association) Ethics and Welfare Group [76].

The owner often sees premature euthanasia as the only option by believing that it is the only way to show care or compassion for his/her dog. In this case, in our opinion, euthanasia is not a form of compassionate care. In other words, empathic atonement to the blind animal may sometimes, but not always, oblige the owner to conduct compassionate euthanasia. Among other things, the vision’s loss produces emotional distress in owners. The inability of many of them to cope with the circumstances surrounding a blind dog’s visual status needs to be considered when communicating medical information to these individuals. Moreover, when advice concerning treatment is given, it would be hopeful that the owner understands all those factors that he/she perceives as important to improve the life quality of his/her blind dog. Although the main objective of medical or surgical treatment should be to improve visual function, this should be understood given those factors important to each owner. To appreciate the severity of the condition that a dog is experiencing and how this is affecting his/her ability to enjoy life, health-related quality of life assessment tools should be carried out as shown by several authors [77,78,79,80]. The scale of Villalbos [81] represents a straightforward means for veterinarians and owners to assess dogs’ quality of life. This tool would help to monitor objectively the progression of the animal’s condition and agree whether euthanasia is necessary and ethically appropriate.

The owner could assist his/her animal gone blind by developing in himself/herself qualities of character such as patience and compassion.

In other words, euthanizing a companion animal occurs because humans cannot bear the costs, financial or emotional, of continued medical care or because the death might save them from more suffering. In some cases, some options may prolong and improve their lives. Several aspects must be considered concerning the good health of the animal, his/her age, medical prospects, and possible alternatives. Given the strong bond between the pet owner and his/her animal, the veterinarian must also attend to the emotional needs of the owner as a part of the provision of a professional business service [82]. The veterinarian should educate the owners on the welfare needs of their animal, including euthanasia as a valid treatment choice by engaging “in a process of ethical reasoning when considering treatment choice” [65,83]. To assist in the euthanasia decision, it could be appropriate to apply the checklist of questions proposed by Edney (1989) [50].

Is the animal in pain, distress, or discomfort?

-Can the animal walk and balance?-Can the animal eat and drink?-Does the animal have inoperable tumors that can cause pain, distress, and discomfort?-Can the animal breathe without difficulty?-Can the animal urinate and defecate normally?-Can the owner cope physically and emotionally with nursing?

In conclusion, owner education with appropriate information and support is required to help him/her better manage his/her animal’s condition [84] (Box 1) because many dogs can live a fulfilled and happy life without their sight.

Box 1Suggestions for the owner.KEY POINTS:
Creating a safe space to retreat might help the dog feel more confident.Assigning a room or a corner and filling it with his/her water and food bowls, favorite toys, and his/her bed. For example, utilizing noisy toys, which can be especially rewarding for the dog.Removing any hazards or sharp objects that might injure him/her.Avoid shift things to prevent confusion and accidents.Talking to the dog can help him/her feel at ease and help locate the owner.Use the voice to get the dog’s attention before stroking him/her to avoid scaring.Installing baby gates at the top and bottom of stairs to keep the dog safe.Using different textured rugs, carpets and flooring might help him/her build a mental map of his/her surroundings.Putting bells on other pets’ collars or shoes might also help the dog locate his/her owner around the house.


It is also important to educate the pet owner of his/her duties to his/her dog and ensure he/she clearly understands any legal implications if he fails to fulfill his/her responsibilities. Therefore, ophthalmologist veterinarians have a crucial role in improving the welfare of both blind dogs and their owners. Requests of killing without a “reasonable reason” are considered morally undesirable and an offense in animal welfare law [63] in European Union where the legislation recognizes animals as sentient beings.

## Figures and Tables

**Table 1 animals-12-00913-t001:** The main causes that lead to blindness. Adapted from Martin, C.L, 2001 [3].

**Opacification of the Clear Media**	**Keratitis**	- Keratoconjunctivitis sicca - Immune-mediated - Multiple ulcerative insults
	**Keratopathy**	- Lipid dystrophies - Endothelial dystrophy degeneration
	**Aqueous turbidity**	fibrin, lipid, hemorrhage (trauma, blood dyscrasias, ocular neovascularization)
	**Cataract**	Genetic, metabolic, toxic, nutritional, radiation, and traumatic
	**Vitreous**	- Haemorrhage (blood dyscrasias, trauma, retinal disease) - Inflammation - Fibrous scars
**Diseases of the Retina**	**Retinopathies**	- Genetic: PRA, central retinal atrophy, Inborn errors of metabolism (IEM) - Nutritional: vitamin E deficiency - Glaucoma - SARDS - Toxic: ivermectin, enrofloxacin
	**Retinal detachment syndromes**	- Genetic: collie eye anomaly (CEA), retinal dysplasia syndromes, Australian shepherd anomaly - Exudative: systemic mycoses, protothecosis, ehrlichiosis, toxoplasmosis, immune-mediated choroiditis - Transudative: hypertension, IV solution overloading - Neoplastic: metastatic, multicentric
	**Chorioretinitis**	- Distemper - Systemic mycoses - Brucellosis - Toxoplasmosis - Ehrlichiosis - Granulomatous meningoencephalitis
**Lesions of the Conducting Mechanism (Optic Nerve, Chiasm, Tracts, and Radiations)**	- Hypoplasia of optic nerves - Granulomatous meningoencephalitis - Distemper encephalitis - Systemic mycoses - Neoplasia involving the chiasm - Traumatic avulsion - Vitamin A deficiency	
**Lesions of the Occipital Cortex**	- Hydrocephalus - Cerebral malformations - Distemper encephalomyelitis - Systemic mycoses - Granulomatous meningoencephalomyelitis - Hepatic encephalopathy - Inborn errors of metabolism (IEM) - Hypoxia - Vascular infarcts - Traumatic edema and/or hemorrhage	

## Data Availability

The data presented in this study are available on request from the corresponding author.

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
