# Peer review of "Animal Welfare Considerations and Ethical Dilemmas Inherent in the Euthanasia of Blind Canine Patients"

_animals, 2022, doi:10.3390/ani12070913_

Round 1
Reviewer 1 Report
Frankly speaking, the paper is understandable overall. However, there is a content divergence between the process of diagnosing blindness and the process of evaluating the choice of euthanasia. The cause is since blindness is not classified more clearly. To make the content of the paper more understandable, the complete blindness, partial blindness, or any of them should be clearly stated in evaluating euthanasia choices. There are some dogs kept without being euthanasia even if they are completely blind. There are some dogs who were euthanized even if they were able to see to some extent.
I comment on each section below.
- In introduction, the authors state “permanently blind” as a meaning of irereversible blindness, but they don’t explain the extent of blindness. It should be shown that there are three groups of euthanized animals:dogs with complete blindness, dogs with partial blindness, dogs with any of them.
- Table 1 indicates the cause of blindness, but it doesn’t show the content of euthanasia choices. Therefore, the authors should not need any photos and they should delete them all.
- In blindness and animal welfare (the blindness from the animal welfare point of view), the authors state a night blindness as a partial blindness category. But they describe that night blindness is a condition with potential pain. Potential pain is unclear wording, I think. Night blindness is a condition with emotional or psychological distress.
- In the same section of blindness and animal welfare, they state the severe ocular pain including some conditions such as endophthalmitis, glaucoma, SARDS, and penetrating ocular trauma. SARDS (sudden acquired retinal degeneration syndrome) is not associated with severe ocular pain. The reason is that sensory receptors are not present in the retina.
- In Scenario 1 (Different scenarios concerning the euthanasia request), the authors state that irreversible blindness requires euthanasia. In dog’s adaptation, there are some dogs who were euthanized even if they were able to see to some extent. The complete blindness, partial blindness, or any of them should be clearly stated in the adaptation to clarify the content of irreversible blindness.
I think such an ethical paper in animals is necessary.
Please hesitate my comment.
Thank you
Author Response
Dear Reviewer,
Thank you very much for your time and all your comments. We have revised the references and increased them. The manuscript has been strongly reviewed following your comments and suggestion.
We thank you for precise and thoughtful comments and constructive criticism, which has led to a better manuscript.
We revised the manuscript considering the suggestions and more detailed answers are given below.
The changes made in the manuscript to address comments are marked up using the
“Track Changes” function.
Frankly speaking, the paper is understandable overall. However, there is a content divergence between the process of diagnosing blindness and the process of evaluating the choice of euthanasia. The cause is since blindness is not classified more clearly. To make the content of the paper more understandable, the complete blindness, partial blindness, or any of them should be clearly stated in evaluating euthanasia choices. There are some dogs kept without being euthanasia even if they are completely blind. There are some dogs who were euthanized even if they were able to see to some extent.
I comment on each section below.
- In introduction, the authors state “permanently blind” as a meaning of irreversible blindness, but they don’t explain the extent of blindness. It should be shown that there are three groups of euthanized animals:dogs with complete blindness, dogs with partial blindness, dogs with any of them.
We have preferred to add a sentence in order to clarify this concept, specifying that the owners may be required euthanasia in animals with a treatable or non-fatal ocular condition.
- Table 1 indicates the cause of blindness, but it doesn’t show the content of euthanasia choices. Therefore, the authors should not need any photos and they should delete them all.
Table 1 reports the main causes of blindness to introduce the reader for understanding the topic. We have deleted the photos
- In blindness and animal welfare (the blindness from the animal welfare point of view), the authors state a night blindness as a partial blindness category. But they describe that night blindness is a condition with potential pain. Potential pain is unclear wording, I think. Night blindness is a condition with emotional or psychological distress.
Probably we misrepresented the concept. We have deleted the example.
- In the same section of blindness and animal welfare, they state the severe ocular pain including some conditions such as endophthalmitis, glaucoma, SARDS, and penetrating ocular trauma. SARDS (sudden acquired retinal degeneration syndrome) is not associated with severe ocular pain. The reason is that sensory receptors are not present in the retina.
We thank the reviewer for this observation. Certainly, SARDS is not a painful condition. We have deleted it.
- In Scenario 1 (Different scenarios concerning the euthanasia request), the authors state that irreversible blindness requires euthanasia. In dog’s adaptation, there are some dogs who were euthanized even if they were able to see to some extent. The complete blindness, partial blindness, or any of them should be clearly stated in the adaptation to clarify the content of irreversible blindness.
We affirm that often owners of dogs with blindness require euthanasia (as observed during our clinical practice) because they don’t know that dogs may have an excellent capacity for adaptation.
I think such an ethical paper in animals is necessary.

Reviewer 2 Report
Review:
Euthanasia in daily clinical practice: Is appropriate to euthanize a dog with blindness diagnosis
Summary: In this article the question about the appropriateness of euthanising dogs with a blindness diagnosis is raised in the perspective of the veterinarian. A number of relevant criteria like presence of pain, onset of blindness or time to adapt are mentioned. The attitude of the veterinarian in particular his duty to explain the difference of dogs and humans is stresses.
Relevance: I think the manuscript deals with an important aspect: euthanasia in a specific clinical context.
Critique: Overall I think many ethical terms are used in a very vague way or imprecisely. I miss a clear and structured ethical evaluation. The English needs some revisions.
More in detail:
• Title: Instead of “Is appropriate” it should be “is it appropriate”. Furthermore, I think the content of the paper is more about an “when is it appropriate….”
• Simple summary: what is meant with “natural” life? Do you think that most of our dogs have a “natural” life?
• Ethical dilemmas: here and throughout the paper the “ethical dilemmas” appear several times, but I am missing a clear definition and explanation of why you think it is an ethical dilemma or you even use it in plural? It looks like you use the term interchangeably with a difficult ethical situation – which is not a precise and correct usage of this term.
• In the abstract you start speaking about companion animals, otherwise it is about dogs.
• I am not sure if the veterinarian can be or should be seen as the “guarantor” of the welfare – I think this assumptions dismisses the patient-owner-vet triad which has been described as the fundamental problem of veterinary medicine.
• What is meant by biodiversity between humans and dogs?
• Introduction: The first part is merely text book knowledge on a number of etiologies, but I am missing the link to the ethical question at hand, any insights if all these disease conditions have been weighted equally. Later in the text some apparently relevant criteria like pain, or available time to adaptation, but I miss a structured thorough ethical analysis.
• L.55 “identifying the cause – do you mean a cause like a specific etiology or the affected structure like retina? If you have the cause as primary cause or reason in mind, then following your argument there would be a problem for all idiopathic cases.
• L.73 what is meant by a “normal” life. For example brachycephalic breeds might have a predisposition for certain eye disease, so for these dogs it might be normal to have eye problems, like keratitis or ulcers. But considering this condition as “normal” might ethically not be advisable.
• L74. Here you mention “ethical dilemmas” (even in plural), but apparently you see the solution (to inform and educate the owner) – so then it is not really a dilemma?
• L. 81: how did you generate the scenarios? Interviews, thought experiments, surveys? Why just four and not more? Typically for a scenario, I would have expected some description like: “Lucy is 10 years old Irish setter and suffers from irreversible blindness due to…” and then have a specific case and later in an ethical assessment about the underlying conflict, the moral theories/relevant ethical criteria.
• LO.116 “Unlike science, ethics must…” I struggle with this statement as ethics is also a scientific discipline. Presumably you mean empirical science vs ethics as a qualitative/normative approach (humanities/social sciences)?
• L.113 “A significant part of welfare is ethics….” – I don’t agree as to become an expert in animal welfare you may study veterinary medicine or biology, but this would not make you an ethicist. Was the idea here with referring to ethics to relate to animal ethics, veterinary medical ethics or applied ethics?
• Please check the references, the num bers do not always correspond to the numbers in the ref list. And for example the ref from Persson does not appear in the main text.
Author Response
Dear Reviewer,
Thank you very much for your time and all your comments. We have revised the references and increased them. The manuscript has been strongly reviewed following your comments and suggestion.
We thank you for precise and thoughtful comments and constructive criticism, which has led to a better manuscript.
We revised the manuscript considering the suggestions and more detailed answers are given below.
The changes made in the manuscript to address comments are marked up using the
“Track Changes” function.
Euthanasia in daily clinical practice: Is appropriate to euthanize a dog with blindness diagnosis
Summary: In this article the question about the appropriateness of euthanising dogs with a blindness diagnosis is raised in the perspective of the veterinarian. A number of relevant criteria like presence of pain, onset of blindness or time to adapt are mentioned. The attitude of the veterinarian in particular his duty to explain the difference of dogs and humans is stresses.
Relevance: I think the manuscript deals with an important aspect: euthanasia in a specific clinical context.
Critique: Overall I think many ethical terms are used in a very vague way or imprecisely. I miss a clear and structured ethical evaluation. The English needs some revisions.
More in detail:
- Title: Instead of “Is appropriate” it should be “is it appropriate”. Furthermore, I think the content of the paper is more about an “when is it appropriate....”
We thank the reviewer for his/her suggestion. We have added “when” and “it”.
- Simple summary: what is meant with “natural” life? Do you think that most of our dogs have a “natural” life? Natural life refers to a dog's physical lifetime or life span. A dog can live around 10-14 years on average(some may naturally live longer, others e.g. may be predisposed to certain diseases that can limit their lifespan).
- Ethical dilemmas: here and throughout the paper the “ethical dilemmas” appear several times, but I am missing a clear definition and explanation of why you think it is an ethical dilemma or you even use it in plural? It looks like you use the term interchangeably with a difficult ethical situation – which is not a precise and correct usage of this term.
We believe that ethics are the moral standards and principles by which entities (individuals and professional organizations) govern their behaviors and decision-making. When these standards and principles conflict with each other in a decision-making situation (such as that in the euthanasia of an animal not seriously ill), an ethical dilemma may occur. Consequently, ethical dilemmas arise when there are situations in which there is a difficult choice to make in a manner that is consistent with accepted ethical guidelines.
- In the abstract you start speaking about companion animals, otherwise it is about dogs.
We have deleted animals and replaced it with dog
- I am not sure if the veterinarian can be or should be seen as the “guarantor” of the welfare – I think this assumption dismisses the patient-owner-vet triad which has been described as the fundamental problem of veterinary medicine.
We thank the reviewer for this observation. We think that veterinarians have central to animal welfare. They have the pragmatism to examine and assess the welfare of animals in their environment and to make recommendations to improve welfare where that is required. For these reasons, we believe that they are guarantors. For example, Petitclerc (2013) (The role of veterinarians in the farm-to-fork food chain and the underlying legal framework. Rev Sci Tech. 2013 Aug;32(2):359-69, 347-58) reports that veterinarians are the guarantors of animal health and protectors of animal resources, providing a vital component of food security and public safety. Based on that, we have underlined that the veterinarian should be seen as the guarantor of animal welfare. However, for better comprehension, we have deleted the sentence.
- What is meant by biodiversity between humans and dogs?
It refers to the diversity of species. We have deleted the sentence.
- Introduction: The first part is merely text book knowledge on a number of etiologies, but I am missing the link to the ethical question at hand, any insights if all these disease conditions have been weighted equally. Later in the text some apparently relevant criteria like pain, or available time to adaptation, but I miss a structured thorough ethical analysis.
Thank you for this comment. The question is whether or not it is permissible to euthanize an animal with blindness and, in our opinion, it is not correct to put down a dog when an appropriate alternative exists and he/she lives a good life, in health; it depends on the dog’s quality of life (a loss of eyesight certainly doesn't mean a poor quality of life). It is acceptable to euthanize a dog when he/she is suffering. We don’t believe that the link to the ethical question is missing. Certainly, not all ocular diseases have equal weight. And for this reason, we have inserted a scenario concerning Central nervous system (CNS) blindness.
- L.55 “identifying the cause – do you mean a cause like a specific etiology or the affected structure like retina? If you have the cause as primary cause or reason in mind, then following your argument there would be a problem for all idiopathic cases.
Identifying the cause means understanding because a dog can go blind (for several reasons, from disease to old age). We have clarified the sentence.
- L.73 what is meant by a “normal” life. For example brachycephalic breeds might have a predisposition for certain eye disease, so for these dogs it might be normal to have eye problems, like keratitis or ulcers. But considering this condition as “normal” might ethically not be advisable.
We have specified above the meaning of a normal life. However, we have deleted “normal” and replaced it with “good”.
We thank the reviewer for the example that he/she has reported. We well know that brachycephalic breeds have a predisposition for certain ocular abnormalities and these problems are constant, but this does not mean that the eye of these breeds is normal. They are conditions due to the typical facial and orbital appearance of brachycephalic dogs, characterized by round skull shapes and plane orbits, leading to a physiological degree of exophthalmia. This makes the eyes more prone to chronic exposure, resulting in a wide variety of problems, such as trauma, exposure keratopathy, superficial pigmentary keratitis, corneal erosion, or even ulceration, etc. They can be medically treated or corrected through surgery with better quality of the patient’s vision. When not treated, they lead to permanent damage which consequently can cause a decrease in the dogs’ visual acuity. This does not mean that the dog must be put down.
- L74. Here you mention “ethical dilemmas” (even in plural), but apparently you see the solution (to inform and educate the owner) – so then it is not really a dilemma?
Indeed, an ethical dilemma does not offer an obvious solution that would comply with ethics al norms, especially when any choice violates the societal and ethical standards. To better understand, we have rephrased the sentence.
- L. 81: how did you generate the scenarios? Interviews, thought experiments, surveys? Why just four and not more? Typically for a scenario, I would have expected some description like: “Lucy is 10 years old Irish setter and suffers from irreversible blindness due to...” and then have a specific case and later in an ethical assessment about the underlying conflict, the moral theories/relevant ethical criteria.
The scenarios are hypothetical and realistic. About your suggestions, we have carried out a description of patient scenarios.
- LO.116 “Unlike science, ethics must...” I struggle with this statement as ethics is also a scientific discipline. Presumably you mean empirical science vs ethics as a qualitative/normative approach (humanities/social sciences)?
We thank the reviewer to have underlined that ethics is also a scientific discipline. We agree. We have re-formulated the sentence.
- L.113 “A significant part of welfare is ethics....” – I don’t agree as to become an expert in animal welfare you may study veterinary medicine or biology, but this would not make you an ethicist. Was the idea here with referring to ethics to relate to animal ethics, veterinary medical ethics or applied ethics?
We refer to veterinary medical ethics.
- Please check the references, the numbers do not always correspond to the numbers in the ref list. And for example, the ref from Persson does not appear in the main text.
We have carefully rechecked all references.

Reviewer 3 Report
EUTHANASIA IN DAILY CLINICAL PRACTICE: IS APPROPRIATE TO EUTHANIZE A DOG WITH BLINDNESS DIAGNOSIS?
The aim of this article was to discuss the suitability of euthanasia in blind dogs. To better assess factors influencing the choice of euthanasia, four different scenarios were constructed describing different situations in the animal aptitude, pet owner, and veterinarian relations. The Authors
underline how the diagnosis of blindness in dogs can sometimes lead to an inappropriate request for euthanasia.
The topic provides useful interesting considerations on the relationship between clinical evaluation and ethics in animal welfare.
Line 37
The authors report many causes of blindness, but it could also be useful to include eye proptosis as a cause of blindness. I have often seen dogs with bilateral proptosis with owners requiring euthanasia even before detecting whether any clinical improvement in vision would have been evident on clinical examination.
Line 48
The authors classify blindness in complete bilateral blindness and incomplete blindness, but it is not clear whether the following discussion will focus on complete blindness (as I believe it is, and I believe is the only reason why the might owner request euthanasia) or even on more less severe vision changes.
Please modify the introduction so that it is clear that the topic refers to complete bilateral blindness or other situations.
Line 114
Delete the
Lines 104-107
Basically I agree with the authors. However, I believe that, from a practical point of view, it is not always easy for the veterinarian to detect a positive emotional state of the dog if not through a calm and peaceful conversation with the owner. Do the authors have any suggestions for setting up such an interview with the owner?
Line 133
“…and, untreated pain…” Delete the comma
Line 333
…”logic of care; moral…” Delete ;
Line 337
Are you sure that the reference is 74? I ‘m quiet sure that the right reference number is 77.
Please check all the reference numbers along the text
Line 371
In the text of the article you refer very often to the owner “education”.
Since, from my point of view, an article must also provide daily clinic indications, could the authors also indicate how the owner could be educated on the welfare needs of his/her animal ? What do they suggest in the veterinary practice?
References
Please check all the reference numbers along the text
Line 517
Replace Associtaion with Association
Author Response
Dear Reviewer,
Thank you very much for your time and all your comments. We have revised the references and increased them. The manuscript has been strongly reviewed following your comments and suggestion.
We thank you for precise and thoughtful comments and constructive criticism, which has led to a better manuscript.
We revised the manuscript considering the suggestions and more detailed answers are given below.
The changes made in the manuscript to address comments are marked up using the
“Track Changes” function.
The aim of this article was to discuss the suitability of euthanasia in blind dogs. To better assess factors influencing the choice of euthanasia, four different scenarios were constructed describing different situations in the animal aptitude, pet owner, and veterinarian relations. The Authors underline how the diagnosis of blindness in dogs can sometimes lead to an inappropriate request for euthanasia.
The topic provides useful interesting considerations on the relationship between clinical evaluation and ethics in animal welfare.
Line 37
The authors report many causes of blindness, but it could also be useful to include eye proptosis as a cause of blindness. I have often seen dogs with bilateral proptosis with owners requiring euthanasia even before detecting whether any clinical improvement in vision would have been evident on clinical examination.
I thank the reviewer for his/her observation. Certainly, one of the blindness may include eye proptosis, also known as exophthalmos. It could result from a wide variety of pathologies, but because in the dog it is most commonly caused by a traumatic event (such as a dog-on-dog fight, blunt force injury or iatrogenic trauma from excessive physical restraint), we have preferred to insert the word “trauma” (line 32). However, considering that the eyes of brachycephalic dogs can develop a proptosis, we have added Brachycephalic Ocular Syndrome.
Line 48
The authors classify blindness incomplete bilateral blindness and incomplete blindness, but it is not clear whether the following discussion will focus on complete blindness (as I believe it is, and I believe is the only reason why the might owner request euthanasia) or even on less severe vision changes.
Please modify the introduction so that it is clear that the topic refers to complete bilateral blindness or other situations.
We refer to complete blindness. In any case, at the end of the introduction (the aim of the study) we have added complete blindness
Line 114: Delete the
Done
Lines 104-107: Basically I agree with the authors. However, I believe that, from a practical point of view, it is not always easy for the veterinarian to detect a positive emotional state of the dog if not through a calm and peaceful conversation with the owner. Do the authors have any suggestions for setting up such an interview with the owner?
A conversation with the owner is useful, but we believe that a veterinarian must know how to evaluate animal welfare through i.e. behavioral indicators. In conclusions we underline that “The veterinarian should educate the owners on the welfare needs of their animal” (lines 367-368).
Line 133: “…and, untreated pain…” Delete the comma
Done
Line 333: …”logic of care; moral…” Delete ;
Done
Line 337: Are you sure that the reference is 74? I ‘m quiet sure that the right reference number is 77. Please check all the reference numbers along with the text
We have changed the reference and carefully rechecked all references.
Line 371: In the text of the article you refer very often to the owner “education”.
Since, from my point of view, an article must also provide daily clinic indications, could the authors also indicate how the owner could be educated on the welfare needs of his/her animal ? What do they suggest in the veterinary practice?
We have added the checklist proposed by Edney (1989) and box 1 (line 425)
References. Please check all the reference numbers along with the text
We have carefully rechecked all references
Line 517
Replace Associtaion with Association
Done

Reviewer 4 Report
Review
Euthanasia in daily clinical practice: Is appropriate to euthanize a dog with blindness diagnosis?
General
The authors provide a comprehensive review of the ethical dilemmas inherent in the decision-making around euthanasia of the blind canine patient.
The subject matter is fascinating, and I have thoroughly enjoyed ruminating over the arguments presented in this paper. The manuscript has, however, been badly let down by poor sentence structures. The poor grammar throughout this manuscript often distracts the reader from the important points being made.
Essentially this paper represents a discussion and literature review on the ethical considerations of convenience euthanasia in animals. This does not just apply to dogs with blindness. I appreciate that visual impairment in dogs is a good pathological condition to consider when discussing the ethics inherent in euthanasia (as it is often a treatable and non-fatal condition) but the arguments presented around convenience euthanasia apply to companion animals with many pathological conditions other than blindness in dogs. The authors may wish to include scenarios that involve pathological conditions in other domestic species.
For this paper to be published, I feel it needs substantial revisions. These revisions should include:
- The title is poorly constructed and needs to reflect that it is the ethics surrounding euthanasia decision-making (in any companion animal species) that is being discussed.
- The aims need to be made clearer. Currently the authors state that the aim was ‘to understand if euthanasia in blind dogs is appropriate or not’. This is too vague as essentially this paper represents an ethical discussion on animal welfare and the biocentric approach to animal euthanasia.
- All references to causes of canine blindness need to be removed. This paper is not about the medicine of canine ophthalmic conditions.
- The paper should be expanded to include a discussion on euthanasia in other companion animal species. Less emphasis should be placed on canine blindness. Perhaps more discussion on animal welfare and the ethics of euthanasia (especially convenience euthanasia) should be included.
- The Discussion and Conclusions need to be combined and re-labelled as a ‘Summary’ or ‘Final Thoughts’
- The grammar and sentence construction need to be improved throughout the paper.
- The Simple Summary and Abstract need to be modified and re-worded once the other changes have been made. For example, the sentence: ‘Described situations varying in the animal aptitude, pet owner, and veterinarian relations’ (Lines 15-16) – makes no sense.
I have provided more detailed feedback below, but this does not cover all the revisions required in this paper.
Introduction
The authors emphasise the importance of the veterinarian adhering strictly to biocentric principles when considering euthanasia. It is important, however, that the authors also discuss the many other factors that need to be considered when deciding whether euthanasia is appropriate. (e.g. suitability of the dog’s domestic environment, size of dog, owner’s ability to manage a blind dog, owner’s ability to pay the surgical or ongoing treatment costs). These are discussed later in the paper but should be introduced in the Introduction
Lines 35-36: Change to: ‘…by certain cancers (brain and orbital tumours), neurologic diseases, and drug overdoses that affect the retina (e.g. an overdose with ivermectin) [1].’
Lines 38-47: Table 1 and Figures 1-4 are unnecessary and should be excluded. This is not a study on the causes of blindness in dogs. The paper is reporting on the conflict between veterinarians and owners about euthanasia decisions concerning a pet with a treatable or non-fatal condition (e.g. blindness). The Introduction should, therefore, be more concerned with the welfare implications of companion animals with non-fatal conditions (e.g. dogs with blindness) and should include analysis of the previous literature related to this.
Line 72-74: The sentence: ‘Despite many dogs with blindness diagnosis can live normal life [12], the owners, distressed for the condition of their animals, require euthanasia.’ - is poorly constructed and implies that the distressed owners require euthanasia! Please re-word.
Lines 74-78: The authors assert that there is an ethical dilemma over euthanasia decisions, for dog blindness, based on biocentric principles. A biocentric viewpoint does not mean that the pet’s ‘interests are considered a priority’. It just means that the client’s opinions do not automatically override the animal’s rights. Decisions over the euthanasia of a pet must be made considering the severity of the condition, the long-term prognosis, the animal’s rights and the interests of the clients. The veterinarian must weigh up all these considerations before deciding whether euthanasia is the best course of action. As the authors state later in the paper, client education is the key to help resolve this ethical dilemma. All these concepts should, therefore, be introduced in the Introduction.
So for Lines 79-80: The sentence should read: ‘To understand if euthanasia in blind dogs is appropriate or not, a study on this topic was undertaken, taking into consideration the interests and needs of the animal as well as all the other factors involved in euthanasia decision-making.’
Line 78: I do not see how reference 13 is related to the sentences that precede it. This must be a mistake.
Line 87: Remove the word ‘Despite’.
Line 97: Remove this last part of the sentence: ‘…in simple terms, it is equivalent to having a good life’.
Lines 107-108: By saying: ‘As in our case, blindness modifies or prevents the perception of the environment.’ – this implies that the authors are blind! Instead say: ‘A medical condition such as blindness can, however, modify animal behaviour as it may modify or prevent the animal’s perception of the environment.’
Line 109 Do not put ‘Of course…’ as this implies that the sentence following this is obvious to the reader.
Lines 113-121: I suspect that the authors’ meanings are being distorted by poor sentence structure.
The definition of ethics provided by the authors is clumsy: ‘Ethics is the process of determining which course of action has the best moral reasons for being undertaken’. In the context of this study, ethics would refer to the moral principles that govern the veterinarian’s behaviour when dealing with a blind patient. Ethics would, therefore, prescribe the veterinarian’s course of action in terms of the blind animal’s welfare as well as the rights/obligations/welfare of the owner.
Lines 116-117: The authors need to explain what is meant by: ‘Unlike science, ethics must be considered when addressing value-based questions such as the acceptability of animal quality of life.’
Lines 122-178: Good points have been raised by the authors regarding pain associated with blindness and thus justifying the argument for bilateral enucleation based on animal welfare. In the next section the authors admit that the owner’s welfare must also be considered. The issue of affordability needs to be addressed here (and not just discussed in one of the later scenarios). Owners might not be able to afford the surgery, ongoing treatment or certain environmental changes required to accommodate a blind dog (especially a large breed).
Lines 222-225: Please change: ‘Though the literature suggests that blind companion animals may have an excellent capacity for adaptation, the owners usually do not appreciate this, and they - distressed because their dogs present abruptly with irreversible blindness - require euthanasia.’
This once again implies that the owners require euthanasia!
Lines 239-240: I’m not sure that reference to the paediatrician model is entirely applicable here. While I accept that a biocentric approach is often adopted by paediatricians (who represent patients with no voice or input – as with animal patients), fortunately euthanasia is never a topic that has to be considered in paediatric medicine.
In addition, references 58 and 59 do not relate to this topic and shed no light on this argument. Please carefully check your references as they do not seem to always align with the text.
Lines 236-242: I feel that the authors’ assertion that vision loss alone is not enough justification for euthanasia is too simplistic. I accept that the veterinarian’s obligation is to fully inform the owner that their perception of blindness may be different to their pet’s perception and that the animal may be able to adapt very easily to a loss of vision. I feel, however, that many other factors need to be considered when deciding on the appropriateness of euthanasia.
Lines 244-255: The argument applies to any animal facing unnecessary euthanasia, not just working dogs with an onset of blindness. Convenience euthanasia remains an ethical dilemma that must be carefully managed by the veterinarian. The veterinarian must act as the advocate for the animal but must also preserve the client-veterinarian relationship. One of your references: Rathwell-Deault et al. (2017) addresses this issue well.
Lines 272-273: The statement: ‘Nevertheless, some veterinarians prioritize the owner’s interest and proceed with euthanasia [60]’ – is too simplistic. Veterinarians face this dilemma every day. As mentioned above, the reasons for euthanasia are often multi-factorial and not just an unprofessional veterinarian pandering to the whims of a callous owner.
The authors actually make this point themselves when they state that: ‘The veterinarian, where there is no legal framework about euthanasia in dogs and cats, is morally obligated to attempt influencing client decisions, although himself/herself he/she also has a duty to the owner and should consider his/her autonomy in decision making [67]. It must be the veterinarian’s responsibility to use his/her judgment on different options valuable for beneficial to the patient and present all available options to the owner ranging from the best practice (gold standard) [68-69] to euthanasia.’ (Lines: 296-301)
One could, however, argue that ‘best practice (gold standard)’ may be opting for euthanasia in some cases.
Lines 286-287: Regarding the statement: ‘Under the legacy of deontology, premature euthanasia cannot be chosen for reasons of convenience.’ - I am not so sure that I agree. Deontological ethics would be based on the duties and rights of the owner and would place value on the intentions of the owner (rather than the outcomes of any action). So, theoretically, if the owner’s intention is sound (e.g. they cannot justify spending all their money on ‘cataract extraction surgery by phacoemulsification’ on their dog as the money is needed to feed their children), then deontological theory would dictate that the intention of the owner justifies the dog’s euthanasia (the outcome).
Discussion and Conclusions
The Discussion does not really wrap up the paper. The Discussion should serve to bring the threads of the arguments and findings from the paper together and to discuss the implications of the findings on future euthanasia decision-making. It would be more suitable to combine the Discussion and Conclusions together within a Summary. The Summary should also contain final recommendations.
Lines 356-359: The authors end the discussion with the statement: ‘There is no list of necessary conditions that hold always and must be met to determine whether a euthanasia decision is acceptable.’
Firstly, the sentence is poorly constructed and should instead be: ‘A list of sacrosanct conditions, that should always be met to justify a euthanasia decision, does not exist’.
Secondly, this final sentence contradicts what the authors have stated earlier in the Discussion in Lines 353-355
Lines 374-375: The authors make the statement: ‘Requests of killing without a “reasonable reason” are considered morally undesirable and an offense in animal welfare law [62]’ - While this is unquestionably true, legislation protecting animals’ right to life (or legislation that recognises animals as sentient beings) differs from country to country.
Author Response
Dear Reviewer,
Thank you very much for your time and all your comments. We have revised the references and increased them. The manuscript has been strongly reviewed following your comments and suggestion.
We thank you for precise and thoughtful comments and constructive criticism, which has led to a better manuscript.
We revised the manuscript considering the suggestions and more detailed answers are given below.
The changes made in the manuscript to address comments are marked up using the
“Track Changes” function.
General
The authors provide a comprehensive review of the ethical dilemmas inherent in the decision-making around euthanasia of the blind canine patient.
The subject matter is fascinating, and I have thoroughly enjoyed ruminating over the arguments presented in this paper. The manuscript has, however, been badly let down by poor sentence structures. The poor grammar throughout this manuscript often distracts the reader from the important points being made.
Essentially this paper represents a discussion and literature review on the ethical considerations of convenience euthanasia in animals. This does not just apply to dogs with blindness. I appreciate that visual impairment in dogs is a good pathological condition to consider when discussing the ethics inherent in euthanasia (as it is often a treatable and non-fatal condition) but the arguments presented around convenience euthanasia apply to companion animals with many pathological conditions other than blindness in dogs. The authors may wish to include scenarios that involve pathological conditions in other domestic species.
We thank the reviewer for his/her considerations. However, we are not agree with these considerations because, in our experience during daily clinical practice, owners required often the euthanasia of their dogs affected from ocular pathologies causing blindness. But these are different point views.
For this paper to be published, I feel it needs substantial revisions. These revisions should include:
- The title is poorly constructed and needs to reflect that it is the ethics surrounding euthanasia decision-making (in any companion animal species) that is being discussed.
Given that this is a re-submission of the paper and a previous reviewer suggested us a similar title, we have preferred propose again it. We have re-worded a part of it on the basis of the suggestion of the reviewer 1.
- The aims need to be made clearer. Currently the authors state that the aim was ‘to understand if euthanasia in blind dogs is appropriate or not’. This is too vague as essentially this paper represents an ethical discussion on animal welfare and the biocentric approach to animal euthanasia.
We have rephrased the aims at the end of introduction.
- All references to causes of canine blindness need to be removed. This paper is not about the medicine of canine ophthalmic conditions.
We are disagree, because it is a paper addressed to veterinary practitioners that must recognize the main causes of blindness in the dog.
- The paper should be expanded to include a discussion on euthanasia in other companion animal species. Less emphasis should be placed on canine blindness. Perhaps more discussion on animal welfare and the ethics of euthanasia (especially convenience euthanasia) should be included.
We don’t understand because we should take into consideration other companion animal if we have only wanted to focus on the dog. For example, if we also consider the cat, in this animal is necessary to carry out other considerations related to ocular diseases in this species and to its behavioral repertoire.
- The Discussion and Conclusions need to be combined and re-labelled as a ‘Summary’ or ‘Final Thoughts’
Done
- The grammar and sentence construction need to be improved throughout the paper.
The paper is been reviewed by a native English speaker
- The Simple Summary and Abstract need to be modified and re-worded once the other changes have been made. For example, the sentence: ‘Described situations varying in the animal aptitude, pet owner, and veterinarian relations’ (Lines 15-16) – makes no sense.
Done
I have provided more detailed feedback below, but this does not cover all the revisions required in this paper.
Introduction
The authors emphasise the importance of the veterinarian adhering strictly to biocentric principles when considering euthanasia. It is important, however, that the authors also discuss the many other factors that need to be considered when deciding whether euthanasia is appropriate. (e.g. suitability of the dog’s domestic environment, size of dog, owner’s ability to manage a blind dog, owner’s ability to pay the surgical or ongoing treatment costs). These are discussed later in the paper but should be introduced in the Introduction
Lines 35-36: Change to: ‘…by certain cancers (brain and orbital tumours), neurologic diseases, and drug overdoses that affect the retina (e.g. an overdose with ivermectin) [1].’
Done
Lines 38-47: Table 1 and Figures 1-4 are unnecessary and should be excluded. This is not a study on the causes of blindness in dogs. The paper is reporting on the conflict between veterinarians and owners about euthanasia decisions concerning a pet with a treatable or non-fatal condition (e.g. blindness). The Introduction should, therefore, be more concerned with the welfare implications of companion animals with non-fatal conditions (e.g. dogs with blindness) and should include analysis of the previous literature related to this.
Thank you for this consideration, but there is a specific paragraph about this topic (2. Blindness from the animal welfare point of view) and, for these reasons we don’t believe necessary to take into consideration in the introduction the welfare implications of companion animals with non-fatal conditions. We have introduced the most common causes of vision loss in dogs to understand how decisions over euthanasia in dogs with blindness could be made considering the severity of the condition/disease and the long-term prognosis.
Line 72-74: The sentence: ‘Despite many dogs with blindness diagnosis can live normal life [12], the owners, distressed for the condition of their animals, require euthanasia.’ - is poorly constructed and implies that the distressed owners require euthanasia! Please re-word.
We have re-worded it.
Lines 74-78: The authors assert that there is an ethical dilemma over euthanasia decisions, for dog blindness, based on biocentric principles. A biocentric viewpoint does not mean that the pet’s ‘interests are considered a priority’. It just means that the client’s opinions do not automatically override the animal’s rights. Decisions over the euthanasia of a pet must be made considering the severity of the condition, the long-term prognosis, the animal’s rights, and, the interests of the clients. The veterinarian must weigh up all these considerations before deciding whether euthanasia is the best course of action. As the authors state later in the paper, client education is the key to helping resolve this ethical dilemma. All these concepts should, therefore, be introduced in the Introduction.
Thank the reviewer for this suggestion. We have added these concepts at the end of the introduction.
So for Lines 79-80: The sentence should read: ‘To understand if euthanasia in blind dogs is appropriate or not, a study on this topic was undertaken, taking into consideration the interests and needs of the animal as well as all the other factors involved in euthanasia decision-making.’
Done
Line 78: I do not see how reference 13 is related to the sentences that precede it. This must be a mistake.
Thank. There were mistakes. We have carefully rechecked all references
Line 87: Remove the word ‘Despite’.
Done
Line 97: Remove this last part of the sentence: ‘…in simple terms, it is equivalent to having a good life’.
Done
Lines 107-108: By saying: ‘As in our case, blindness modifies or prevents the perception of the environment.’ – this implies that the authors are blind! Instead say: ‘A medical condition such as blindness can, however, modify animal behaviour as it may modify or prevent the animal’s perception of the environment.’
Done
Line 109 Do not put ‘Of course…’ as this implies that the sentence following this is obvious to the reader.
Done
Lines 113-121: I suspect that the authors’ meanings are being distorted by poor sentence structure.
The definition of ethics provided by the authors is clumsy: ‘Ethics is the process of determining which course of action has the best moral reasons for being undertaken’. In the context of this study, ethics would refer to the moral principles that govern the veterinarian’s behaviour when dealing with a blind patient. Ethics would, therefore, prescribe the veterinarian’s course of action in terms of the blind animal’s welfare as well as the rights/obligations/welfare of the owner.
We have inserted this concept.
Lines 116-117: The authors need to explain what is meant by: ‘Unlike science, ethics must be considered when addressing value-based questions such as the acceptability of animal quality of life.’
The sentence structure was poor. We have re-worded it.
Lines 122-178: Good points have been raised by the authors regarding pain associated with blindness and thus justifying the argument for bilateral enucleation based on animal welfare. In the next section the authors admit that the owner’s welfare must also be considered. The issue of affordability needs to be addressed here (and not just discussed in one of the later scenarios). Owners might not be able to afford the surgery, ongoing treatment or certain environmental changes required to accommodate a blind dog (especially a large breed).
In our opinion, points raised are discussed in the Scenario 3. Financial constraints of the pet owner.
We have preferred not examine their in section 2.2. because we have discussed the animal welfare problems connected with blindness
Lines 222-225: Please change: ‘Though the literature suggests that blind companion animals may have an excellent capacity for adaptation, the owners usually do not appreciate this, and they - distressed because their dogs present abruptly with irreversible blindness - require euthanasia.’
This once again implies that the owners require euthanasia!
The sentence has been modified.
Lines 239-240: I’m not sure that reference to the paediatrician model is entirely applicable here. While I accept that a biocentric approach is often adopted by paediatricians (who represent patients with no voice or input – as with animal patients), fortunately euthanasia is never a topic that has to be considered in paediatric medicine.
We have deleted the sentence.
In addition, references 58 and 59 do not relate to this topic and shed no light on this argument. Please carefully check your references as they do not seem to always align with the text.
Thank. There were mistakes. We have carefully rechecked all references
Lines 236-242: I feel that the authors’ assertion that vision loss alone is not enough justification for euthanasia is too simplistic. I accept that the veterinarian’s obligation is to fully inform the owner that their perception of blindness may be different to their pet’s perception and that the animal may be able to adapt very easily to a loss of vision. I feel, however, that many other factors need to be considered when deciding on the appropriateness of euthanasia.
We agree with this affirmation. In fact, few lines after (…) we underline that “deciding to euthanize a blind dog may become permissible if the animal is geriatric with concomitant debilitating diseases and little hope of full recovery”.
Lines 244-255: The argument applies to any animal facing unnecessary euthanasia, not just working dogs with an onset of blindness. Convenience euthanasia remains an ethical dilemma that must be carefully managed by the veterinarian. The veterinarian must act as the advocate for the animal but must also preserve the client-veterinarian relationship. One of your references: Rathwell-Deault et al. (2017) addresses this issue well.
We thank the reviewer for the consideration. Certainly, some veterinarians in the convenience euthanasia choice give priority to respecting the owner’s interest.
Lines 272-273: The statement: ‘Nevertheless, some veterinarians prioritize the owner’s interest and proceed with euthanasia [60]’ – is too simplistic. Veterinarians face this dilemma every day. As mentioned above, the reasons for euthanasia are often multi-factorial and not just an unprofessional veterinarian pandering to the whims of a callous owner.
The authors actually make this point themselves when they state that: ‘The veterinarian, where there is no legal framework about euthanasia in dogs and cats, is morally obligated to attempt influencing client decisions, although himself/herself he/she also has a duty to the owner and should consider his/her autonomy in decision making [67]. It must be the veterinarian’s responsibility to use his/her judgment on different options valuable for beneficial to the patient and present all available options to the owner ranging from the best practice (gold standard) [68-69] to euthanasia.’ (Lines: 296-301).
We have corrected it
One could, however, argue that ‘best practice (gold standard)’ may be opting for euthanasia in some cases.
We have inserted this concept
Lines 286-287: Regarding the statement: ‘Under the legacy of deontology, premature euthanasia cannot be chosen for reasons of convenience.’ - I am not so sure that I agree. Deontological ethics would be based on the duties and rights of the owner and would place value on the intentions of the owner (rather than the outcomes of any action). So, theoretically, if the owner’s intention is sound (e.g. they cannot justify spending all their money on ‘cataract extraction surgery by phacoemulsification’ on their dog as the money is needed to feed their children), then the deontological theory would dictate that the intention of the owner justifies the dog’s euthanasia (the outcome).
We do not agree for two reasons:
1) relating to the five needs/freedoms in which any person responsible for an animal must fulfill, there is "To be protected from pain, suffering, injury and disease". Consequently, the pet owner if does not treat him/her fails to protect the animal from suffering not complying with these guidelines
2) In Italy, the Guide to Professional Conduct for Veterinarian states that dogs may only be euthanized when they are 'seriously or incurably ill or proven to be dangerous'.
Discussion and Conclusions
The Discussion does not really wrap up the paper. The Discussion should serve to bring the threads of the arguments and findings from the paper together and to discuss the implications of the findings on future euthanasia decision-making. It would be more suitable to combine the Discussion and Conclusions together within a Summary. The Summary should also contain final recommendations.
We thank you for this suggestion. We have combined the two parts.
Lines 356-359: The authors end the discussion with the statement: ‘There is no list of necessary conditions that hold always and must be met to determine whether a euthanasia decision is acceptable.’
Firstly, the sentence is poorly constructed and should instead be: ‘A list of sacrosanct conditions, that should always be met to justify a euthanasia decision, does not exist’.
Secondly, this final sentence contradicts what the authors have stated earlier in the Discussion in Lines 353-355.
We have deleted the sentence.
Lines 374-375: The authors make the statement: ‘Requests of killing without a “reasonable reason” are considered morally undesirable and an offense in animal welfare law [62]’ - While this is unquestionably true, legislation protecting animals’ right to life (or legislation that recognizes animals as sentient beings) differs from country to country.
Certainly, the legislation that recognizes animals as sentient beings is different from country to country. We have added where.
We have carefully rechecked all references
Line 517
Replace Associtaion with Association
Done

Round 2
Reviewer 1 Report
I will accept your last manuscript for publication.
The authors modified more simple and more practical documents for us rather than the original paper. I think your last manuscript may be given a contribution in the ethics of the small animal ophthalmology.
Author Response
Thank you for your suggestion.
Reviewer 2 Report
Please refer to the attached doc.

Author Response
Dear Reviewer,
Thank you very much for your time and all your comments.
We revised the manuscript considering the suggestions and more detailed answers are given below.
The changes made in the manuscript to address comments are marked up using the
“Track Changes” function.
Review: Euthanasia in daily clinical practice: When is it appropriate to put down a dog with a blindness diagnosis (revised version) Thanks for considering my comments. I still feel that the manuscript would benefit from describing precisely what you mean and including ethical terms correctly and with caution.
- Abstract natural life : If you refer here to the typical life span, possibly breed-specific, please be precise and write something like reaching a similar age compared to non-affected dogs or describing in other words that there is no need for a premature death. Later in the manuscript you refer to Quality of life , which I think is more relevant than the mere length of life (see the discussion of zoo animals animals in captivity may live much longer than wild animals, but here solely the length of a life does not automatically mean that they have a good life).
- R. We thank for the suggestion. We have rephrased the sentence
- Ethical dilemmas (abstract): I am still concerned as I think, what you describe is not an ethical dilemma in a strict sense. I miss a clear and referenced definition of the term ethical dilemma One example for an ethical dilemma from Morgan: A moral dilemma, in a strict sense, is a conflict between responsibilities or obligations of exactly equal moral weight. In a wider sense, moral dilemmas occur when there are competing responsibilities with no obvious way to prioritize one responsibility over others. doi:10.1016/j.cvsm.2006.09.008 So in your argumentation, describing the vet as an advocate for the animal or as a pediatrician, it is without (ethical) ambiguity that for a blind animal with a good prognosis, the vet is to let the animal alive (and for an animal with constant pain and a bad prognosis, the vet would presumeably act in the interest of the animal to euthanise it). Of course, it can be extremely hard, if the vet is confronted with an unjustified wish of the owners to euthanise (convenience euthanasia). The vet himself knows what would be the right course of conduct (let the animal alive), but he/she simply cannot do it, because the owner wants to euthanise. So I question (in the definition from Morgan), if the obligation towards the owner has exactly the same moral weight than the obligation for a vet towards an animal? This would be a prerequisite of being an ethical dilemma following this definition. I think the responsibility/obligation of the vet towards the animal is more important. In this line, it is not an ethical dilemma, because the vet (once all the aspects related to diagnosis, duration, etc are known), knows the right thing to do (not euthanise), but cannot do it due to an utional constraint wner). This is a definition of moral distress which a vet may experience. Or another definition An ethical dilemma may be defined as an ethical situation in which one has a choice between equally unsatisfactory alternatives. ver you do you will harm someone (the animal or the owner). But in the case of a blind dog with a good prognosis, who stays alive, would you really think that the owners are harmed? It is a satisfactory alternative, that the dog can live. An example of an ethical dilemma could be a vet involved in disease control of a highly contagious and fast spreading disease. So on one hand, infected animals should be culled as fast as possible, to stop the spread and protect so far unaffected animals from contact with the pathogen. On the other hand, each animal should be killed without pain, suffering carefully killing takes time. So here, whatever the vet does, rushing to kill as many animals as possible or assuring that each animal is killed without pain, he/she will do harm to either the infected animals to be culled or the so far not infected animals which then would have a higher chance to get the infection. I think what is missing is a link to what was named by Tannenbaum as the fundamental problem of veterinary professional ethics relating to the vet-patient-client triad . As vet we want to treat the animal, but are paid by the owner. This raises the ethical difficult situation. What your paper adds is to support vets in decision-making and counselling in cases of blind dogs by referring to the checklist and the keypoints and to highlight that nowadays blindness does not necessarily leads to euthanasia. Thus, this also hopefully reduces potential moral uncertainty (this is also not the same as an ethical dilemma) for the vet. Please check carefully the parts where speak of ethical dilemmas (summary and L. 76). The latter one is more about moral uncertainty, i.e. not knowing what to do because there are conflicting moral principles.Because the veterinarian should also consider the owner’s autonomy in decision making, we believe that this is an ethical dilemma. As underlined by an other reviewer “a balanced argument between the welfare of the animal and the rights/obligations/welfare of the owner must be considered. The veterinarian must act as the advocate for the animal but must also preserve the client-veterinarian relationship”
- R. We discuss this situation in lines 341-350. In lines 12 and 76 we have replaced ethical dilemma with conflicting moral principles
- English: please check the wording by a native speaker (which I am not).
For example
210 applied instead of applies
L-304 omit was
R. Done.
- Related to scenario 1, I think here it is not just the blindness which is relevant for the decision, but also the cushing (incl. its etiology) and possibly clinical manifestations like frequent urinations.
R. We think that this dog not should be euthanized because if treated (pharmacologically and surgically) the prognosis could be good. An alternative to euthanasia could be adoption. We have inserted some considerations.
- Related to scenario 2, I am missing the consideration of the capacity of the owner to support a blind dog. One might argue if the owner was not able to correctly perform the treatment, would he then be able to support a blind dog? Would it not be better for this specific dog to be euthanised to avoid future pain and negligence? Which reasons of the owner to euthanize would you consider to be morally acceptable and why?
- R.We have better discussed this case
- Vets as of animal welfare . My aim was not to delete this sentence, but to critically discuss it. Of course vets are the professionals with the expertise to assess the welfare and propose/conduct measures to assure/improve the welfare. But, they are not the only ones to decide, referring again the vet-patient-owner triad, and if the owner wants to euthananize, (in most cases) the vets would lack power to guarantee animal welfare. I think assuming that vets have to guarantee the welfare while there might be situations where they simply don the power to do so (if no rehoming is possible, lacking financial ressources) puts an enormous pressure on vets. Vets do have a high risk of suicide. My assumption is that this tendency of feeling responsible (while one actually lacks in some situations the power to act in favor of the animals) contributes to moral distress and compassion fatigue/burnout. In this vein, vets should guarantee a good clinical examination and a good counselling which hopefully would lead to a good welfare for the animal, but they cannot guarantee the welfare of the animal.
- R. Thank you for the explanation
- L.116 I don with this sentence, mostly for methodological reasons. I think rather that Animal welfare science is not interchangeable with or a synonym of animal ethics for several reasons» as discussed in the second paragraph of the introduction in https://doi.org/10.3390/ani12050586
R. We share the opinion of Rollin. However, we have added your thoughts and that of other Authors

Reviewer 4 Report
Euthanasia in daily clinical practice: When is it appropriate to put down a dog with a blindness diagnosis?
The sentence structure and grammar remain poor. This manuscript requires extensive re-wording to bring it up to the standards required of an international journal.
For example, sentences such as: ‘Despite many dogs with blindness diagnosis can live a good life [13], the owners, require euthanasia of the animal became blind but with a treatable and/or non-fatal condition (“convenience euthanasia”), either for financial reasons or simply because they do not want, he/she anymore.’ (Lines 66-69) …are very poorly constructed. This is one of many examples.
While this manuscript offers a comprehensive review of the ethical dilemmas inherent in the decision-making around euthanasia of the blind canine patient, I still maintain that this paper represents an ethical discussion on the animal welfare implications of convenience euthanasia. While the authors have addressed the ethical and animal welfare implications involved in convenience euthanasia in dogs with blindness, the authors need to acknowledge that the arguments presented apply to all companion animals suffering from treatable or non-terminal conditions.
While I appreciate that the authors are following advice from a fellow reviewer, I feel that the revised title still does not fully capture the manuscripts’ intention. The paper represents a comprehensive review of the animal welfare considerations and ethical dilemmas inherent in the decision-making around euthanasia of the blind canine patient. It is not just discussing when it is appropriate to euthanise a blind dog. The title should instead be along the lines of:
A review of the animal welfare considerations and ethical dilemmas inherent in the euthanasia of the blind canine patient
Lines 74-76: Lacking a clearly defined set of rules when approaching a canine blindness case is not an ethical dilemma but could result in the development of an ethical dilemma.
Lines 83-85: The aims are still unclear. Surely the aim of this paper is to discuss the ethical dilemmas inherent in the decision-making around euthanasia of the blind canine patient (as well as the animal welfare implications) and not just: ‘To better assess factors influencing euthanasia choice...’
Line 200: Change ‘For this reason, as will be mentioned in the discussion section…’ as there is no longer a Discussion section.
The authors have included my explanation that ethics prescribes the veterinarian’s course of action in terms of the blind animal’s welfare as well as the rights/obligations/welfare of the owner. They have not, however, expanded much on the rights/obligations/welfare of the owner. The authors do mention later that the veterinarian has a duty to the owner and should consider the owner’s autonomy in decision making, but this must be made more explicit throughout the manuscript. Clearly a balanced argument between the welfare of the animal and the rights/obligations/welfare of the owner must be considered. The veterinarian must act as the advocate for the animal but must also preserve the client-veterinarian relationship.
I still maintain that the authors’ assertion that: ‘Under the legacy of deontology, premature euthanasia cannot be chosen for reasons of convenience.’…represents a flawed assumption. The obligation of an owner to fulfil the animal’s five needs/freedoms is unquestionable, but this is not dictated by the ‘legacy of deontology’. Perhaps it would be better to not refer to the term ‘deontology’ and instead emphasise that convenience euthanasia is illegal in Italy and other countries within the European Union.
Author Response
Dear Reviewer,
Thank you very much for your time and all your comments.
We revised the manuscript considering the suggestions and more detailed answers are given below.
The changes made in the manuscript to address comments are marked up using the
“Track Changes” function.
Euthanasia in daily clinical practice: When is it appropriate to put down a dog with a blindness diagnosis?
The sentence structure and grammar remain poor. This manuscript requires extensive re-wording to bring it up to the standards required of an international journal.
For example, sentences such as: ‘Despite many dogs with blindness diagnosis can live a good life [13], the owners, require euthanasia of the animal became blind but with a treatable and/or non-fatal condition (“convenience euthanasia”), either for financial reasons or simply because they do not want, he/she anymore.’ (Lines 66-69) …are very poorly constructed. This is one of many examples.
While this manuscript offers a comprehensive review of the ethical dilemmas inherent in the decision-making around euthanasia of the blind canine patient, I still maintain that this paper represents an ethical discussion on the animal welfare implications of convenience euthanasia. While the authors have addressed the ethical and animal welfare implications involved in convenience euthanasia in dogs with blindness, the authors need to acknowledge that the arguments presented apply to all companion animals suffering from treatable or non-terminal conditions.
- While I appreciate that the authors are following advice from a fellow reviewer, I feel that the revised title still does not fully capture the manuscripts’ intention. The paper represents a comprehensive review of the animal welfare considerations and ethical dilemmas inherent in the decision-making around euthanasia of the blind canine patient. It is not just discussing when it is appropriate to euthanise a blind dog. The title should instead be along the lines of:
A review of the animal welfare considerations and ethical dilemmas inherent in the euthanasia of the blind canine patient
Thank you for your suggestion. We have modified the title.
- Lines 74-76: Lacking a clearly defined set of rules when approaching a canine blindness case is not an ethical dilemma but could result in the development of an ethical dilemma.
We have modified the sentence.
- Lines 83-85: The aims are still unclear. Surely the aim of this paper is to discuss the ethical dilemmas inherent in the decision-making around euthanasia of the blind canine patient (as well as the animal welfare implications) and not just: ‘To better assess factors influencing euthanasia choice...’
We have modified the sentence.
- Line 200: Change ‘For this reason, as will be mentioned in the discussion section…’ as there is no longer a Discussion section.
We have changed the discussion section with the final thoughts section.
- The authors have included my explanation that ethics prescribes the veterinarian’s course of action in terms of the blind animal’s welfare as well as the rights/obligations/welfare of the owner. They have not, however, expanded much on the rights/obligations/welfare of the owner. The authors do mention later that the veterinarian has a duty to the owner and should consider the owner’s autonomy in decision making, but this must be made more explicit throughout the manuscript. Clearly a balanced argument between the welfare of the animal and the rights/obligations/welfare of the owner must be considered. The veterinarian must act as the advocate for the animal but must also preserve the client-veterinarian relationship.
We have considered and clarified the owner’s autonomy in decision-making.
- I still maintain that the authors’ assertion that: ‘Under the legacy of deontology, premature euthanasia cannot be chosen for reasons of convenience.’…represents a flawed assumption.The obligation of an owner to fulfil the animal’s five needs/freedoms is unquestionable, but this is not dictated by the ‘legacy of deontology’. Perhaps it would be better to not refer to the term ‘deontology’ and instead emphasise that convenience euthanasia is illegal in Italy and other countries within the European Union.
We have deleted the sentence. We cannot affirm that the convenience euthanasia is illegal in EU because the situation is different in every European country.
